# Benchmarking and Analyzing 3D-aware Image Synthesis with a Modularized Codebase

Qiuyu Wang[1]    Zifan Shi[2]    Kecheng Zheng[1]    Yinghao Xu[3]    Sida Peng[4]    Yujun Shen[1]

[1]Ant Group        [2]HKUST        [3]CUHK        [4]ZJU

## Abstract

Despite the rapid advance of 3D-aware image synthesis, existing studies usually adopt a mixture of techniques and tricks, leaving it unclear how each part contributes to the final performance in terms of generality. Following the most popular and effective paradigm in this field, which incorporates a neural radiance field (NeRF) into the generator of a generative adversarial network (GAN), we build a well-structured codebase, dubbed *Carver*, through modularizing the generation process. Such a design allows researchers to develop and replace each module independently, and hence offers an opportunity to fairly compare various approaches and recognize their contributions from the module perspective. The reproduction of a range of cutting-edge algorithms demonstrates the availability of our modularized codebase. We also perform a variety of in-depth analyses, such as the comparison across different types of point feature, the necessity of the tailing upsampler in the generator, the reliance on the camera pose prior, *etc.*, which deepen our understanding of existing methods and point out some further directions of the research work. We release code and models here to facilitate the development and evaluation of this field.

## 1   Introduction

Learning a 3D-aware generative model has received growing attention considering its practical applications, such as digital avatar and virtual reality. In addition to image quality and diversity, which are widely pursued by 2D generation, 3D-aware image synthesis also requires the output images to be spatially consistent across different viewing directions. Due to the lack of large-scale 3D data, previous attempts [46, 39, 20, 8, 56, 40, 58] propose to learn the 3D-aware model with 2D images as the only supervision. To accomplish such a challenging task, the most popular solution is to introduce neural radiance fields (NeRFs) into generative adversarial networks (GANs) as the inductive bias. In this way, the GAN generator acquires the awareness of the underlying geometry when rendering an image, whose fidelity is promised by the competition with the discriminator [19].

Towards an effective incorporation between NeRFs and GANs, many techniques have been proposed, such as SIREN activation [6] and tri-plane representation [8]. However, the popularity of this field unintentionally leads to some problems. (1) Existing algorithms are usually developed with different codebases [7, 21, 57, 9], which adopt different 3D coordinate systems and rendering pipelines, making it hard to transfer well-trained models from one codebase to another. (2) State-of-the-art performance is usually achieved through an adequate combination of many techniques and tricks, where some are novel while some are inherited from prior arts. However, existing codebases typically hold an entangled implementation, making it hard to recognize the contribution of each part. (3) A follow-up problem of an entangled implementation is the inconvenience of drawing merits from different approaches, causing additional burden to the advance of this field.

37th Conference on Neural Information Processing Systems (NeurIPS 2023) Track on Datasets and Benchmarks.

Table 1: Analyses of 3D-aware image synthesis performed with our modularized codebase.

| Module | Analysis | Observation |
|---|---|---|
| Point Embedder | MLP *v.s* Volume *v.s* Tri-plane | Different point features exhibit competitive capacities. |
| | Combination of multiple types of point feature | The contribution is marginal compared to a single type of point feature. |
| | Number of planes | Bi-planes performs on par with tri-planes. |
| Feature Decoder | Decoder depth (*i.e.*, number of layers) | The depth only matters for MLP-based point embedder. |
| | Activation function | SIREN is better than ReLU when upsampler module is absent. |
| Volume Renderer | Density-based *v.s* SDF-based | SDF-based representation currently lags behind the density-based one. |
| Upsampler | Effects on the generation quality and consistency | Upsamplers benefit the quality but harm the multi-view consistency. |
| Pose Sampler | Effects of the pre-defined pose priors | The more accurate the poses are, the better the generation quality is. |

This work fills in this gap with a modularized codebase for 3D-aware image synthesis. In particular, we reformulate the generation process into a bunch of modules, as shown in Fig. 1, including a pose sampler, a stochasticity mapper, a point sampler, a point embedder, a feature decoder, a volume renderer [36], and an upsampler. Besides, we also integrate a visualizer and an evaluator into our codebase to facilitate online evaluation. With such a design, we are able to share the 3D coordinate system and the rendering pipeline for all methods, and hence leave the research effort to the improvement of each individual module. In summary, our contributions are three-fold.

- We build a highly-modularized easy-to-use codebase for 3D-aware image synthesis, and also use it to re-implement a range of classic algorithms in this field. The on-par or even better reproduction results suggest that existing approaches can be easily reformulated into a module combination following our pipeline, demonstrating the availability of our toolkit.
- Our codebase allows users to replace a particular module (*e.g.*, from the function perspective) arbitrarily and independently, facilitating the per-module evaluation as well as the design integration from various methods. We believe our codebase could help the community with a more convenient algorithm development.
- Thanks to the modularity, we perform a variety of in-depth analyses regarding different modules, which is beyond the capacity of previous functionally entangled codebases. The studies and observations are summarized in Tab. 1. Besides deepening our understanding of this field, these analyses also help point out some further directions of the research work, as discussed in Sec. 4.3.

## 2 Background

**Task setting.** 3D-aware image synthesis aims at generating multi-view images only from 2D image collections. Basically, it always incorporates 3D inductive bias into 2D generative models, enabling the generation of 3D models from 2D images without any 3D data. Previous efforts always concentrate on 3D shape generation [53, 5, 29, 54], which typically demands extensively annotated 3D datasets for training models. Thanks to the progress of neural implicit fields [36, 42, 35, 32, 18, 1, 2, 44, 51] and generative models [19, 28, 3, 45, 24] especially GANs [25–27], 3D-aware image synthesis has been advanced significantly in terms of the 3D consistency and visual quality.

**Common solution.** Neural Radiance Field (NeRF) [36] $F(x, v) \rightarrow (c, \sigma)$ regresses color $c \in \mathbb{R}^3$ and volume density $\sigma \in \mathbb{R}$ from coordinate $x \in \mathbb{R}^3$ and viewing direction $v \in \mathbb{S}^2$, parameterized with multi-layer perceptron (MLP) networks. Recent attempts on 3D-aware image synthesis propose to condition NeRF with a latent code $z$, resulting in their generative forms like GRAF [46], $G(x, v, z) \rightarrow (c, \sigma)$, to generate multi-view images of the object. Many recent attempts have been made to improve generative NeRF, including the incorporation of convolutional upsamplers [39, 20, 8, 56, 40, 58], hybrid representations [56, 8, 4, 61], other implicit fields [41, 40, 55], patch-wise training [50], and MPI-based rendering [63, 14]. Some works have leveraged alternative 3D representations, such as meshes [17, 30], voxel grids [47, 64, 37, 38, 16, 22], and depth [48], to achieve 3D-aware image synthesis, which is not the primary focus of our paper.

**Datasets.** We benchmark 3D-aware image synthesis on three datasets, including FFHQ [26], ShapeNet Cars [10] and Cats [62]. *FFHQ* [26] is a 2D dataset with $70K$ unique high-quality face images of resolution $1024 \times 1024$. A variety of accessories are also covered in the dataset, including eyeglasses, earrings, *etc.* The face images are shot from frontal views and near-frontal views. *ShapeNet Cars* [10] is the car subset of ShapeNet [10], which consists of $8K$ CAD model. Both the geometry and texture are stored for each model. We random sample camera poses that span

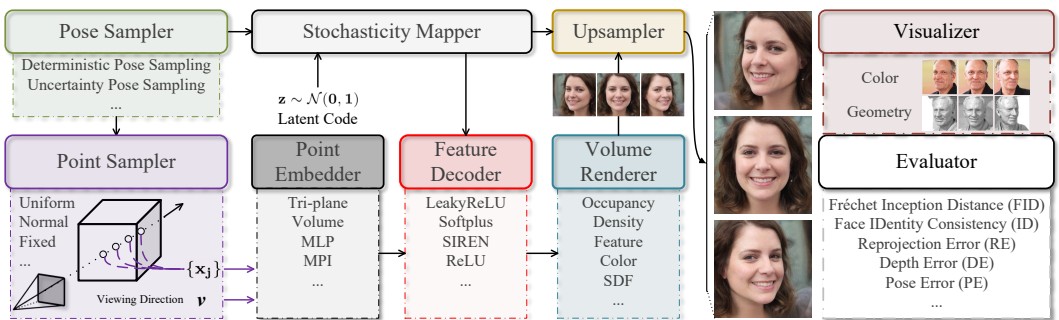

Figure 1: Overview of our modularized pipeline for 3D-aware image synthesis, which modularizes the generation process in a universal way. Each module can be improved independently, facilitating algorithm development. Note that the discriminator is omitted for simplicity.

the entire 360° camera azimuth and 180° camera elevation distributions to render CAD model into 2D images. **Cats** [62] contains 6,444 cat face images of resolution 256×256. The camera poses are generally on the front and near-front of the cat. **Discussion.** Typically, the datasets used for 3D-aware image synthesis comprise single-object datasets with an easy-to-define camera pose prior, as opposed to compositional scene datasets. The objects in these datasets have similar zoom, scale, and geometry, making them suitable for learning 3D representation from 2D observations. Previous studies [6, 46, 20, 56, 8] commonly use these datasets for evaluation, while other datasets, such as CARLA [15], LSUN Bedroom [60], and CelebA [33], are not covered in this work.

**Metrics.** We include Fréchet Inception Distance, reprojection error, face identity consistency, pose error and depth error to evaluate the methods. **Fréchet Inception Distance (FID)** [23] is adopted to evaluate the quality and the diversity of the synthesized images from 3D-aware image synthesis model. **Reprojection Error (RE)** [56] measures the distance between two adjacent views by warping them to each other based on the rendered depth maps. **Identity Consistency (ID)** [8] is designed for facial identity consistency evaluation. For each generated identity, mean Arcface [12] cosine similarity score is calculated between pairs of views rendered from random camera poses. **Pose Error (PE)** [8, 56] measures the accuracy of the rendered objects' poses. A pre-trained pose estimator (*e.g.*, head pose estimator) is leveraged to estimate the pose from the rendered image. Pose error calculates the distance between the estimated pose and the given camera pose for rendering. **Depth Error (DE)** [8] is used to evaluate the quality of the underlying shape in 3D-aware image synthesis, where a pre-trained depth estimator is adopted to predict the depth for the rendered 2D image. The predicted depth is then compared with the rendered depth to indicate the shape quality.

## 3 Modularized pipeline for 3D-aware image synthesis

We propose a modularized framework that employs GANs to generate 3D representations from single-view images. The overall pipeline can be seen in Fig. 1. Our framework aims to achieve modularity in the design, and thus enables easy integration of different components. By providing a modularized platform that grants easy usage and flexible configurations, our framework can foster collaborative efforts toward improving the codebase over the long term. The following sections will provide details of each module.

**Pose sampler.** 3D-aware generative models rely on camera pose $\theta$ to regulate the synthesized view of an object. As a result, pose sampler is proposed to sample poses as the model input during training. The pose sampler in our framework supports two kinds of pose sampling: stochastic pose sampling and deterministic pose sampling. Stochastic pose sampling enables sampling pose from a random or pre-defined pose distribution, *i.e.*, Gaussian distribution or uniform distribution. Deterministic pose sampling allows for sampling poses for each sample with its ground-truth pose. Ground-truth poses are easily accessible for some datasets, such as ShapeNetCars [10]. Although the FFHQ [26] and Cats [62] datasets do not include ground-truth poses, we utilize a readily available face pose estimator [13] to approximate these poses, treating the results as our ground-truth. By offering both types of pose sampling, our framework provides flexibility in pose selection for users.

**Point sampler.** Sampling points is an essential step for NeRF because it requires a set of points along a single ray for rendering. However, various methods use different coordinate systems to sample

points, which can make it challenging to study components from different methods. To alleviate this issue, we develop a unified point sampler that allows users to sample points in a consistent coordinate system. Initially, we sample coordinates in pixel space and then transform them into camera space using camera intrinsics. We then use the sampled camera-to-world matrix (*i.e.*, camera extrinsic) to transform the coordinates into world space. Our point sampler facilitates the combination of components from different methods by providing a simple workflow for point sampling in a unified coordinate system. This simplifies the sampling process, making it easier for users to sample points and integrate components from various methods.

**Stochasticity mapper.** Following the StyleGAN family [26, 27], we learn a stochasticity mapper that takes a random noise $\mathbf{z} \sim \mathcal{N}(0, 1)$ as input, and outputs an intermediate latent code $\mathbf{w} \in \mathcal{W}$ to modulate the styles of 3D-aware synthesis modules (*e.g.*, feature decoder, and upsampler). The learnable latent space $\mathcal{W}$ can better simulate the native distribution of real data. Moreover, we can incorporate camera parameters $\theta$ as conditions into the stochasticity mapper to enhance the 3D consistency of synthesized images. This allows the target view to influence the scene synthesis process, leading to more realistic and accurate 3D-aware synthesized images.

**Point embedder.** The point embedder is responsible for transforming raw point coordinates into point features. Our framework provides several options for this transformation, including extracting from a multi-layer perceptron (MLP), querying from a feature volume, a tri-plane or multiplane image (MPI) representation, or their combinations. By leveraging explicit structure information encoded in a volume or tri-plane representation, the point embedder can provide a more detailed description, leading to a significant impact on the quality of the final rendering. Therefore, the design of the point embedder is of great importance. Our framework offers a unified interface for extracting MLP, volume, tri-plane, and MPI-based point features, providing users with a convenient development experience.

**Feature decoder.** The feature decoder is a module that converts the extracted point features from the point embedder into color, density, or SDF values. Typically, the feature decoder consists of a multi-layer perceptron (MLP) that uses ReLU as an activation function. However, recent research has demonstrated that SIREN [49], which employs periodic activation functions, exhibits greater capability in modeling fine details than ReLU-based representations. Our framework supports both ReLU and SIREN-based MLP decoders, as well as some other choices, providing users with the flexibility to choose the most suitable decoder for their specific application needs.

**Volume renderer.** The volume renderer is a critical component that transforms decoded colors, densities, or other properties into 2D images, making it easy to receive supervision from 2D training datasets. Our framework includes support for the basic integration formula in NeRF [36] and offers a range of clamping modes for color and density values, as well as additional options for volume rendering. These features provide users with the flexibility to customize their rendering process according to their specific needs, enabling them to achieve high-quality results that meet the demands of their particular application.

**Upsampler.** Volumetric rendering can result in a large memory footprint and slow rendering speeds when generating high-resolution images. To maintain efficiency, many recent approaches [40, 20, 56, 8] employ convolutional upsamplers to render high-resolution images. These methods first generate a low-resolution feature map and then use an upsampler to progressively add appearance information and increase the resolution of the rendering. The typical upsampler consists of upsampling layers with $1 \times 1$ or $3 \times 3$ convolutional layers. Our framework provides users with a range of upsamplers [40, 20, 56, 8] to choose from, allowing them to customize their upsampling process to meet their specific needs.

**Evaluator.** The current codebase for 3D GANs lacks systematic evaluation metrics. To address this issue, our codebase includes support for various evaluation metrics for 3D generation, such as FID [23], reprojection error [56], face identity consistency [8], depth error [8], and pose error [8]. We have integrated widely used 3D face reconstruction model [13] and face recognition model [12] into our framework, making it convenient to test various metrics. The inclusion of these metrics enables quantitative evaluation and allows the community to gain a better understanding of the effects of the key components from various methods.

Table 2: Overview of methods supported by our codebase. We provide an outline of the modules in our supported methods and present the reproduced results with our codebase on FFHQ [26], with the FID [23] used as the evaluation metric.

| Method | Pose Sampler | Point Embedder | Feature Decoder | Volume Renderer | Upsampler | Resolution | Official | Reproduction |
|---|---|---|---|---|---|---|---|---|
| GRAF [46] | Stochastic | MLP | ReLU | Density, Color | No | 128×128 | 46.30 | **45.50** |
| $\pi$-GAN [6] | Stochastic | MLP | SIREN | Density, Color | No | 128×128 | 29.90 | **27.81** |
| StyleSDF [40] | Stochastic | MLP | SIREN | SDF, Color, Feature | Yes | 256×256 | 11.50 | **10.96** |
| | | | | | | 512×512 | **10.07** | 10.71 |
| | | | | | | 1024×1024 | **10.01** | 10.14 |
| StyleNeRF [20] | Stochastic | MLP | ReLU | Density, Color, Feature | Yes | 256×256 | **8.00** | 8.31 |
| | | | | | | 512×512 | 7.80 | **7.37** |
| | | | | | | 1024×1024 | 8.10 | **8.08** |
| VolumeGAN [56] | Stochastic | Volume | LeakyReLU | Density, Color, Feature | Yes | 256×256 | **9.10** | 10.37 |
| GRAM [14] | Deterministic | MPI | SIREN | Occupancy, Color | No | 256×256 | 14.50 | **13.83** |
| EpiGRAF [50] | Deterministic | Tri-plane | LeakyReLU | Density, Color | No | 512×512 | 9.92 | **9.19** |
| EG3D [8] | Deterministic | Tri-plane | Softplus | Density, Color, Feature | Yes | 256×256 | 4.80 | **4.72** |
| | | | | | | 512×512 | 4.70 | **4.63** |

**Visualizer.** In addition to quantitative metrics, our framework also supports qualitative visualization of generated images and extraction of the underlying geometry. With our developed codebase, users can easily generate multi-view images, as well as obtain 3D shapes of each generated sample.

# 4 Experiments

This section commences with a concise overview of the implementation details of our experiments. Following this, we review the methods that are supported by our codebase. We then present our observations and analyses of the primary modules in our framework, highlighting which components are essential in 3D GANs. Finally, based on our experimental settings, we discuss promising future directions for 3D-aware image synthesis.

**Implementation details.** We employ our developed codebase to conduct extensive experiments on the main modules in 3D GANs. To ensure a systematic evaluation, we use the state-of-the-art 3D-aware GAN, EG3D [8], as our backbone model and substitute each module with alternative choices. For instance, to investigate the point embedder, we substitute its tri-plane with feature volume or other representations. We benchmark 3D-aware image synthesis on FFHQ [26], Cats [62], and ShapeNet Cars [10] datasets. The models are trained on the FFHQ and Cats datasets at a resolution of 256, and the ShapeNet Cars dataset at a resolution of 128. We train all models on 8 NVIDIA A100 GPUs for 25 million images, with a batch size of 32. Following the approach in EG3D [8], we set the generator learning rate to 0.0025 and the discriminator learning rate to 0.002. More details of our experimental settings can be found in Appendix A.

## 4.1 Supported methods and reproduced results

We support all highly representative models in the field of 3D-aware image synthesis, as they encompass almost all mainstream point embedders, including MLP [46, 6, 40, 20, 14], volume [56], tri-plane [50, 8] and MPI [14]. As for feature decoder activation, they include both SIREN [6, 40, 14], ReLU [46, 20] or some other activations [56, 8, 50]. Additionally, some of these methods include an upsampler [40, 20, 56, 8], while others do not [46, 6, 14, 50]. We then provide a brief introduction to each of these methods, which are supported by our codebase. **GRAF** [46] is the first work that learns a generative model for implicit radiance fields in 3D-aware image synthesis. **$\pi$-GAN** [6] introduces a mapping network to condition layers in the SIREN [49] using feature-wise linear modulation (FiLM) [43], a novel architecture in 3D GANs. **StyleSDF** [40] is another method that incorporates signed distance functions (SDFs) into 3D generative models and achieves impressive results in terms of visual and geometric quality. **StyleNeRF** [20] integrates the neural radiance field (NeRF) [36] into a style-based generator to improve rendering efficiency and 3D consistency for high-resolution image generation. **VolumeGAN** [56] uses a feature volume to represent the underlying geometry, enabling high-fidelity 3D-aware image synthesis. **GRAM** [14] adopts the multiplane image (MPI) representation to constrain point sampling and radiance field learning on 2D manifolds, facilitating

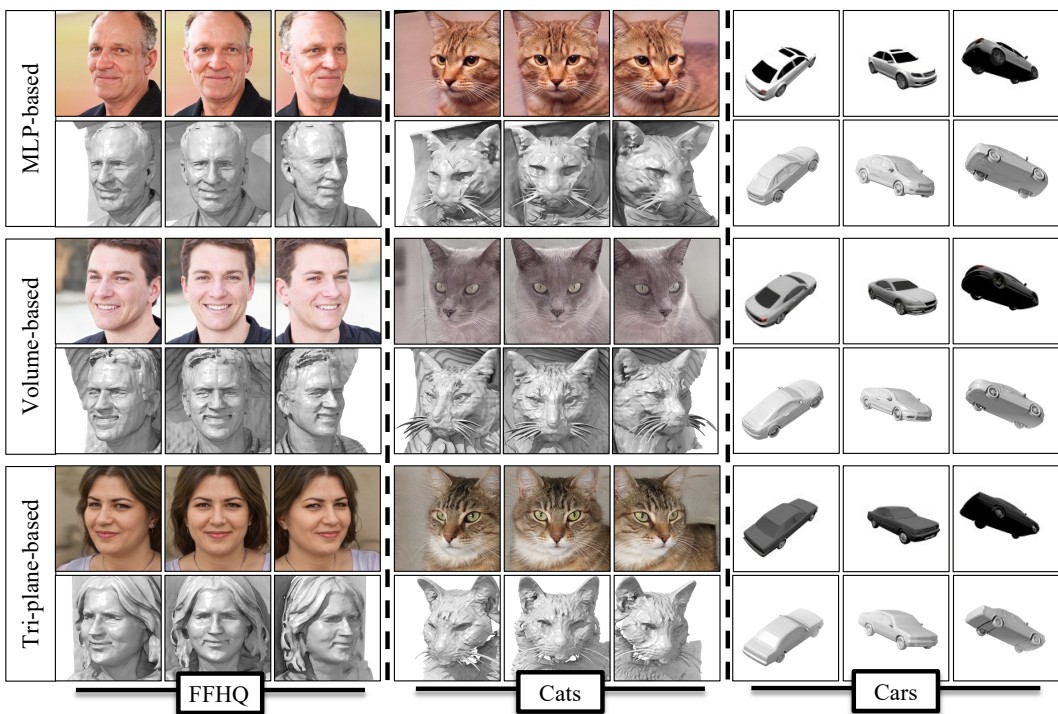

Figure 2: Qualitative comparison across various single point embedders on FFHQ [26], Cats [62] and ShapeNet Cars [10], where the MLP-based, volume-based, and tri-plane-based point features exhibit on-par performance in generating multi-view consistent images and high-quality geometries.

fine-detail learning. **EpiGRAF** [50] proposes a new patch sampling strategy to stabilize training and accelerate convergence. Finally, **EG3D** [8] proposes a tri-plane-based 3D GAN framework that is efficient and expressive for high-resolution geometry-aware image synthesis.

Meanwhile, to evaluate the performance of our implementation, we conduct experiments on the FFHQ dataset, and the results are presented in Tab. 2. Upon comparison of our reproduction with the original implementation, we have observed a similar level of performance in terms of the officially reported metrics. The marginal differences observed in the FID for all methods indicate that our codebase is capable of precisely replicating the official results. The convenience afforded by the shared coordinate system and the versatility of each module has allowed us to retrain various models in a unified and optimized setting, leading to generally improved results in our reproduction compared to the official implementation.

### 4.2 Analyses

**Types of different point embedders.** Based on Tab. 3 and Fig. 2, we can conclude that point features extracted by different point embedders exhibit competitive capacities when combined with the upsampler. The reason may be that the upsampler enhances the modeling ability of scene appearance and relieves the burden of encoding appearance with neural fields in 3D space. Note that tri-plane-based point embedder is more computationally efficient than MLP-based and volume-based point embedders.

**Combination of multiple point embedders.** The results in Tab. 3 and Fig. 3 show that combining the outputs of multiple embedders has a negligible impact on the final outcome. When training the model using multiple embedders, the network tends to find the simplest way to obtain the desired outcome,

Table 3: Analysis of different types of point features.

| Point Embedder | | | FFHQ [26] | | | | | Cats [62] | Cars [10] |
| MLP | Volume | Tri-plane | FID↓ | ID↑ | DE↓ | PE↓ | RE↓ | FID↓ | FID↓ |
|---|---|---|---|---|---|---|---|---|---|
| ✓ | ✗ | ✗ | 5.15 | 0.777 | 0.470 | $5.0e^{-4}$ | 0.091 | 4.05 | 2.42 |
| ✗ | ✓ | ✗ | 4.65 | **0.778** | 0.413 | $5.1e^{-4}$ | **0.085** | **3.59** | **2.25** |
| ✗ | ✗ | ✓ | 4.72 | 0.743 | 0.547 | $\mathbf{4.5e^{-4}}$ | 0.111 | 3.99 | 2.75 |
| ✓ | ✓ | ✗ | 4.70 | 0.773 | **0.334** | $5.1e^{-4}$ | 0.086 | 3.87 | 2.55 |
| ✓ | ✗ | ✓ | 4.69 | 0.748 | 0.465 | $5.3e^{-4}$ | 0.104 | 4.42 | 2.59 |
| ✗ | ✓ | ✓ | 4.68 | 0.735 | 0.378 | $4.6e^{-4}$ | 0.100 | 4.41 | 2.78 |
| ✓ | ✓ | ✓ | **4.62** | 0.769 | 0.467 | $4.7e^{-4}$ | 0.091 | 4.70 | 2.65 |

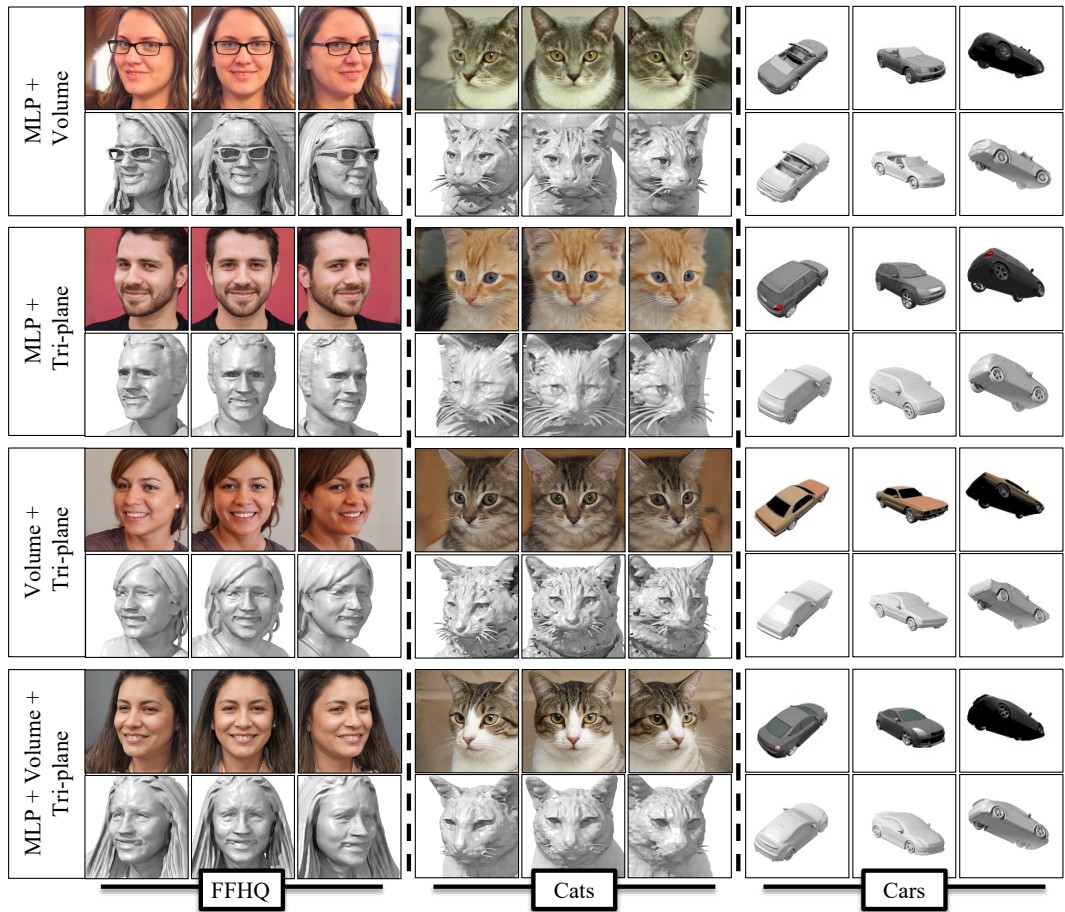

Figure 3: Qualitative comparison across various composite point embedders on FFHQ [26], Cats [62] and ShapeNet Cars [10], where these compound point features exhibit on-par performance in generating multi-view consistent images and high-quality geometries.

which may result in the output features of certain embedders being overlooked. Therefore, using a single type of point feature is sufficient for model training.

**Number of planes.** We also investigate the plane-based point embedders by examining the necessity of using three planes, as in [8]. To accomplish this, we substitute the tri-plane with both single-plane and bi-plane representations. As expected, the single-plane representation performs poorly, as the model cannot

Table 4: Analysis of plane-based point features.

| Point Embedder | FFHQ [26] | | | | | Cats [62] | Cars [10] |
|---|---|---|---|---|---|---|---|
| | FID↓ | ID↑ | DE↓ | PE↓ | RE↓ | FID↓ | FID↓ |
| Bi-plane (XY + XZ) | 4.55 | 0.738 | 0.398 | $4.6e^{-4}$ | **0.083** | 4.33 | 3.21 |
| Bi-plane (XY + ZY) | **4.38** | 0.746 | **0.379** | **$4.4e^{-4}$** | 0.104 | **3.95** | 2.81 |
| Bi-plane (XZ + ZY) | 4.52 | **0.754** | 0.385 | $5.5e^{-4}$ | 0.095 | 4.77 | **2.50** |
| Tri-plane | 4.72 | 0.743 | 0.547 | $4.5e^{-4}$ | 0.111 | 3.99 | 2.75 |

accurately determine a point's location in 3D space with only one plane. However, the bi-plane representation slightly outperforms the tri-plane representation, as shown in Tab. 4. The bi-plane model accurately determines 3D point positions without ambiguities and has fewer parameters. Thus the bi-plane representations are on par with the tri-plane representations in 3D GANs. Nonetheless, this does not necessarily imply that fewer planes are always superior. The datasets we trained on are object-level and relatively simple, making the bi-plane representation sufficient. In more complex tasks, additional planes may be required.

**Depth of feature decoder.** As illustrated in Tab. 5, the depth of the feature decoder plays a crucial role in the accuracy of MLP-based point features. However, for volume and plane-based point features, the depth of the feature decoder appears to be less significant. This can be attributed to the fact that,

in the case of volume or tri-plane-based point features, the feature volume or feature plane predominantly determines the point representation's capability, while the feature decoder serves as a simple mapper to convert the point features into corresponding density or color values. Conversely, for MLP-based point features, the model relies solely on the feature decoder to transform the raw coordinates to the density or color values, and thus the depth of feature decoder determines the model's capacity.

Table 5: Analysis of the depth of feature decoder.

| Point Embedder | Depth | FFHQ [26] | | | | |
|---|---|---|---|---|---|---|
| | | FID↓ | ID↑ | DE↓ | PE↓ | RE↓ |
| MLP | 4 | 17.22 | 0.761 | 0.807 | $12.2e^{-4}$ | 0.105 |
| | 8 | 7.39 | **0.782** | 0.552 | $7.3e^{-4}$ | **0.087** |
| | 16 | **5.15** | 0.777 | **0.470** | $\mathbf{5.0e^{-4}}$ | 0.091 |
| Volume | 4 | 5.65 | 0.784 | 0.437 | $4.4e^{-4}$ | 0.095 |
| | 8 | 5.18 | **0.787** | **0.381** | $\mathbf{4.0e^{-4}}$ | 0.100 |
| | 16 | **4.65** | 0.778 | 0.413 | $5.1e^{-4}$ | **0.085** |
| Tri-plane | 2 | **4.72** | 0.743 | 0.547 | $4.5e^{-4}$ | 0.111 |
| | 4 | 4.77 | 0.750 | **0.414** | $\mathbf{4.4e^{-4}}$ | **0.101** |
| | 8 | 5.58 | **0.750** | 0.566 | $5.6e^{-4}$ | 0.108 |

**Activation type of feature decoder.** As illustrated in Tab. 6, we find that without the upsampler, SIREN-based MLP outperforms the ordinary MLP while with the upsampler the result turns out to be the opposite. One plausible explanation for this observation could be the design of the upsampler module, which may have the tendency to amplify the "ripple" artifacts induced by the SIREN-based layers [6], while mitigating the blurry artifacts produced by the ordinary layers.

Table 6: Analysis of the activation type used in feature decoder.

| Activation Type | FFHQ [26] | | | | |
|---|---|---|---|---|---|
| | FID↓ | ID↑ | DE↓ | PE↓ | RE↓ |
| - *w/ upsampler* | $256 \times 256$ | | | | |
| SIREN | 11.66 | 0.763 | **0.352** | $9.1e^{-4}$ | 0.089 |
| ReLU | **7.39** | **0.782** | 0.552 | $\mathbf{7.3e^{-4}}$ | **0.087** |
| - *w/o upsampler* | $64 \times 64$ | | | | |
| SIREN | **6.58** | **0.741** | **0.340** | $6.6e^{-4}$ | **0.071** |
| ReLU | 7.30 | 0.729 | 0.498 | $\mathbf{4.6e^{-4}}$ | 0.084 |

**Geometric representation.** The results presented in Tab. 7 indicate that the SDF-based representations generally yields inferior quantitative metrics compared to vanilla density-based representations for all point embedders. These findings are consistent with previous works [59]. Moreover, we observe that utilizing sphere initialization and additional regularization terms, such as eikonal loss and minimal surface loss [40], can adversely affect GAN training, leading to flawed results. Thus, incorporating SDF does not offer superior generation quality compared to density-based representations.

Table 7: Analysis of the underlying geometric representation.

| Geometric Representation | | FFHQ [26] | | | | |
|---|---|---|---|---|---|---|
| | | FID↓ | ID↑ | DE↓ | PE↓ | RE↓ |
| MLP | SDF | 8.87 | 0.610 | 0.874 | $5.9e^{-4}$ | 0.184 |
| | Density | **5.15** | **0.777** | **0.470** | $\mathbf{5.0e^{-4}}$ | **0.091** |
| Volume | SDF | 7.27 | 0.676 | 0.938 | $\mathbf{5.0e^{-4}}$ | 0.200 |
| | Density | **4.65** | **0.778** | **0.413** | $5.1e^{-4}$ | **0.085** |
| Tri-plane | SDF | 13.31 | 0.534 | 0.626 | $10.9e^{-4}$ | 0.161 |
| | Density | **4.72** | **0.743** | **0.547** | $\mathbf{4.5e^{-4}}$ | **0.111** |

**Pose priors.** As shown in Tab. 8 and Fig. 4, the pose priors significantly impacts the quality of the generated 3D geometry. Specifically, when training with the random pose distribution (RPD), the generated 3D geometry lacks normal facial structures and exhibits inconsistency in novel view synthesis. On the other hand, when trained solely with the accurate pose distribution (APD), the model tends to overfit to a "flat face shape" and consequently struggles with generating different views of faces. However, when provided with ground-truth pose (GTP) information, the model can produce photo-realistic images and generate adequate and high-quality geometry.

Table 8: Analysis of using different pose priors.

| Pose Prior | FFHQ [26] | | | | |
|---|---|---|---|---|---|
| | FID↓ | ID↑ | DE↓ | PE↓ | RE↓ |
| MLP *w/* RPD | 14.56 | 0.413 | 1.513 | $5.8e^{-2}$ | 0.405 |
| MLP *w/* APD | 9.96 | **0.788** | 1.659 | $5.9e^{-2}$ | 0.407 |
| MLP *w/* GTP | **5.15** | 0.777 | **0.470** | $\mathbf{5.0e^{-4}}$ | **0.091** |
| Volume *w/* RPD | 10.47 | 0.429 | 1.562 | $5.5e^{-2}$ | 0.390 |
| Volume *w/* APD | 7.34 | 0.731 | 1.125 | $4.8e^{-2}$ | 0.367 |
| Volume *w/* GTP | **4.65** | **0.778** | **0.413** | $\mathbf{5.1e^{-4}}$ | **0.085** |
| Tri-plane *w/* RPD | 15.18 | 0.427 | 2.181 | $5.6e^{-2}$ | 0.379 |
| Tri-plane *w/* APD | 5.45 | **0.764** | 1.502 | $5.4e^{-2}$ | 0.405 |
| Tri-plane *w/* GTP | **4.72** | 0.743 | **0.547** | $\mathbf{4.5e^{-4}}$ | **0.111** |

**Upsampler.** As evidenced in Tab. 9, the model incorporating an upsampler exhibits superior image quality, owing to its capacity to enhance image details. Nonetheless, the inclusion of an upsampler, typically accomplished through convolutional

Table 9: Analysis of the effect of upsampler, with tri-plane point feature as an example.

| Upsampler | Resolution | FFHQ [26] | | | | | Training Time | Inference Speed |
|---|---|---|---|---|---|---|---|---|
| | | FID↓ | ID↑ | DE↓ | PE↓ | RE↓ | | |
| ✓ | $256\times256$ | **4.72** | 0.743 | 0.547 | $\mathbf{4.5e^{-4}}$ | 0.111 | **2.7 Days** | **49 FPS** |
| ✗ | $256\times256$ | 6.86 | **0.749** | **0.443** | $6.2e^{-4}$ | **0.104** | 6.9 Days | 20 FPS |

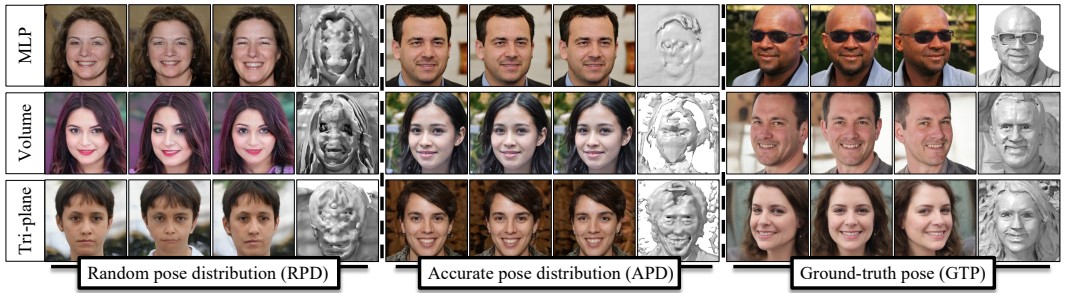

Figure 4: Qualitative comparison across different pose priors, where having ground-truth pose for each training example plays a vital role in helping the model learn adequate geometry.

layers and non-linear activation, may compromise multi-view consistency. Conversely, eliminating the upsampler yields heightened view consistency, albeit at the expense of significantly increased computational cost, manifested in prolonged training and inference times.

### 4.3 Outlooks

Based on the developed codebase, our experimental analysis has revealed numerous unresolved challenges in 3D GANs that warrant further investigation to enhance its applicability in downstream domains. Here, we propose several promising avenues for future research in the realm of 3D-aware image synthesis, with the aim of fostering continued progress and facilitating breakthroughs in this rapidly evolving field.

**Training stability.** Training 3D-aware image synthesis models is often prone to mode collapse, resulting in unsatisfactory image generation and inferior geometry. This may be due to a mismatch between the distribution of physically meaningful factors, such as camera poses and rendering parameters, and that of real images. Therefore, it is crucial to investigate the training stability of 3D-aware image synthesis models.

**Efficiency.** Training 3D-aware image synthesis models can take 3-10 days on multiple high-end GPUs, and the models can be slow during inference, which limits their practicality in downstream applications. Therefore, the need to improve the training efficiency of 3D generative models is increasing, and enhancing the inference efficiency is crucial for their practical deployment in real-world applications.

**Pose acquisition.** The role of pose in 3D GAN training is of great significance, but obtaining accurate pose information from in-the-wild data is a non-trivial task. One potential solution is to incorporate a pose estimator into the 3D GAN model. Further research may focus on improving pose acquisition methods for 3D-aware image synthesis models.

**Hybrid representations.** The current representations employed in 3D GANs are primarily implicit representations. However, in some complicated scenarios, these representations may prove insufficient. For instance, when modeling a human head, the implicit representation is suitable for hair modeling, while explicit representations such as meshes may be more appropriate for face modeling. To address this issue, future research should focus on exploring hybrid representations that can handle more complex data.

## 5   Conclusion

This paper proposes a modularized codebase for 3D-aware image synthesis development, enabling researchers to independently develop and replace each module. This approach provides a means to compare various methods fairly and recognize their key contributions from a module perspective. Extensive experimental analyses reveal the potential future research directions for 3D-aware image synthesis models.

## Broader Impact

The development of 3D-aware image synthesis has significant potential for impact in fields such as entertainment, gaming, and virtual reality. This technology has the ability to revolutionize the way we create and interact with virtual environments by generating realistic 3D models and images. However, as with any emerging technology, ethical considerations must be taken into account to prevent potential misuse for malicious purposes such as the creation of realistic fake images or videos. Therefore, researchers and developers must consider the social and ethical implications of this technology and work collaboratively with stakeholders to develop appropriate guidelines and best practices, ensuring that it is developed and used ethically and responsibly. Open and transparent discussions about the ethical implications of this technology can help maximize its potential benefits while minimizing potential risks.

## Acknowledgments

We would like to express our gratitude to Kai Zhu for his valuable assistance in reproducing EpiGRAF [50] and to Jiaxin Xie for her invaluable help in reproducing GRAM [14]. We extend our thanks to Hao Ouyang and Yuxi Xiao for their insightful comments and discussions, particularly their suggestion on refining this manuscript. We are also thankful to the anonymous reviewers for their valuable feedback and suggestions, which greatly contributed to improving the clarity of this manuscript.

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

# Appendix

## A   Experimental details

### A.1   Backbone

In this study, we adopt the state-of-the-art 3D-aware GAN, EG3D [8], as our backbone model, which we have reproduced in our modularized codebase. This codebase is utilized to perform all experiments presented in this paper. Our investigation involves substituting each module in 3D GANs with alternative choices to study their individual effects. For instance, to explore the point embedder, we replace its tri-plane with feature volume or other representations. Similarly, to investigate the feature decoder, we alter the depth or activation function of the decoder. Given our modularized design, these changes can be easily implemented by adjusting the parameters in the configuration files.

In our experiments, we use only one-stage training to save computational resources. Specifically, we perform the training only at a neural rendering resolution of 64×64, without stepping the resolution up to $128 \times 128$.

### A.2   Point embedder

In this work, we mainly explore the capacity of three point embedders: MLP, volume, and tri-plane, as well as their combinations. The MLP-based point embedder utilizes a multi-layer perceptron (MLP) to transform raw point coordinates into point features. The volume-based point embedder queries point features from the feature volume, while the tri-plane based point embedder queries point features from the tri-plane feature representation.

The experimental settings for the tri-plane based point embedder are identical to those of [8]. For the MLP-based point embedder, we adopt an MLP network to extract point features, similar to [20]. This network employs ReLU activation, 1×1 convolutions modulated by style vectors, and has a depth of 16, with a hidden layer dimension of 128 and an output layer dimension of 64. Notably, for MLP-based point features, the point embedder and the feature decoder are actually the same. As for the volume-based point embedder, we generate the feature volume using a generator that utilizes 3D convolutions, which is the same network architecture as [56]. The feature volume resolution is set as 64×64. The feature decoder for volume-based point features is an MLP network with the same architecture as the MLP-based point embedder. Other settings including geometric representation, upsampler, pose priors, *etc.*, remain consistent with our backbone model.

We also investigate the impact of composite point embedders, which entail combining two or more point embedders. Specifically, we explore combinations of MLP and volume, MLP and tri-plane, volume and tri-plane, and MLP, volume, and tri-plane. The combination of MLP and volume involves concatenating the MLP-based point features and volume-based point features along the feature channel dimension, while the other combinations follow the same principle. To conserve computational resources during training with composite point embedders, we set the MLP-based point features as the raw point coordinates, with the MLP serving as an identical mapping.

## A.3 Feature decoder

Considering that the feature decoder typically comprises a multi-layer perceptron, it is crucial to investigate the impact of its depth and activation layer type on the performance. To this end, we conduct an empirical study on the depth and activation type of the feature decoder. Specifically, we perform experiments on three point features, including MLP, volume, and tri-plane, utilizing the experimental setup inherited from Appendix A.2, but with varying depths of the MLP. In terms of the activation type of feature decoder, we employ both SIREN-based layers and ordinary ReLU-based layers in two settings: with and without an upsampler. And the depth of both MLPs was set to 8, with a hidden dimension of 128 and an output dimension of 64. When training the model with a SIREN-based layer, similar to [6], we set the generator learning rate to 0.00006 and the discriminator learning rate to 0.0002.

## A.4 Geometric representation

Prior works, such as NeuS [52], have incorporated the signed distance function (SDF) into the volume rendering formula to enable the reconstruction of smooth surface models. Moreover, 3D GANs [40] have explored the use of SDF as a geometric representation for consistent generation. Hence, our goal is to examine the effects of two distinct geometric representations: density and SDF. We perform experiments on three point embedders: MLP, volume, and tri-plane. The baseline models utilizing vanilla density are identical to the model described in Appendix A.2. For SDF-based models, the feature decoder outputs SDF values. We also incorporate sphere initialization, eikonal loss, and minimal surface loss during training, with the same loss weight as [40].

## A.5 Upsampler

Our study aims to analyze the influence of the upsampler in 3D GANs. To ensure a fair comparison, we conduct experiments on the generation of 256×256 resolution images. When an upsampler is not utilized, the neural rendering resolution is the same as the image resolution, resulting in significantly longer training time and higher computational costs. We conduct experiments on a model without an upsampler, and disable dual discrimination since the output images are only at a fixed resolution of 256×256. Other settings remain consistent with the backbone model.

## A.6 Pose priors

We additionally explore pose priors on the FFHQ [26] dataset. To obtain the accurate pose distribution (APD) of the dataset, we adopt the approach described in [6], where the pose distribution is modeled as a Gaussian prior. Camera poses are sampled from a normal distribution with a vertical mean of $\pi/2$ radians, standard deviation of 0.155 radians; and a horizontal mean of $\pi/2$ radians, a standard deviation of 0.3 radians. Regarding the random pose distribution (RPD), a Gaussian distribution is also assumed for the pose, with only the vertical/horizontal mean and standard deviation being randomly assigned. And we introduce the acquisition of ground-truth poses in Appendix B.

# B  Dataset details

We conduct our experiments on three datasets, including FFHQ [26], Cats [62], and ShapeNet Cars [10]. Since the original data of these datasets lacks pose labels, we perform preprocessing steps for each dataset. We follow [8] to align, crop and get pose matrix for each image in the FFHQ [26] dataset. Cats [62] contains more than $6K$ real-world cat images, and the data is preprocessed following [14]. The ShapeNet Cars [10] dataset comprises various synthetic car models. We use the dataset rendered from [8] which is composed of approximately $530K$ images. Unlike the forward-facing datasets, its camera poses encompass the full range of $360°$ horizontal and $180°$ vertical distributions. In our experiments, the resolution of $256 \times 256$ is employed for FFHQ and Cats datasets, while for the ShapeNet Cars dataset, a resolution of $128 \times 128$ is used.

Table A1: Training time and inference speed of models with different point embedders. The tri-plane-based point embedder exhibits the highest computational efficiency.

| Point Embedder | | | FFHQ [26] 256×256 | | Cats [62] 256×256 | | Cars [10] 128×128 | |
|---|---|---|---|---|---|---|---|---|
| MLP | Volume | Tri-plane | Training Time | Inference Speed | Training Time | Inference Speed | Training Time | Inference Speed |
| ✓ | ✗ | ✗ | 5.6 Days | 29 FPS | 6.1 Days | 29 FPS | 6.9 Days | 23 FPS |
| ✗ | ✓ | ✗ | 6.8 Days | 24 FPS | 7.3 Days | 24 FPS | 8.3 Days | 20 FPS |
| ✗ | ✗ | ✓ | **2.7 Days** | **49 FPS** | **3.1 Days** | **49 FPS** | **2.7 Days** | **48 FPS** |
| ✓ | ✓ | ✗ | 6.9 Days | 24 FPS | 7.5 Days | 24 FPS | 8.4 Days | 20 FPS |
| ✓ | ✗ | ✓ | 2.8 Days | 49 FPS | 3.3 Days | 49 FPS | 2.8 Days | 48 FPS |
| ✗ | ✓ | ✓ | 3.8 Days | 38 FPS | 4.4 Days | 38 FPS | 3.9 Days | 35 FPS |
| ✓ | ✓ | ✓ | 3.9 Days | 38 FPS | 4.5 Days | 38 FPS | 4.0 Days | 35 FPS |

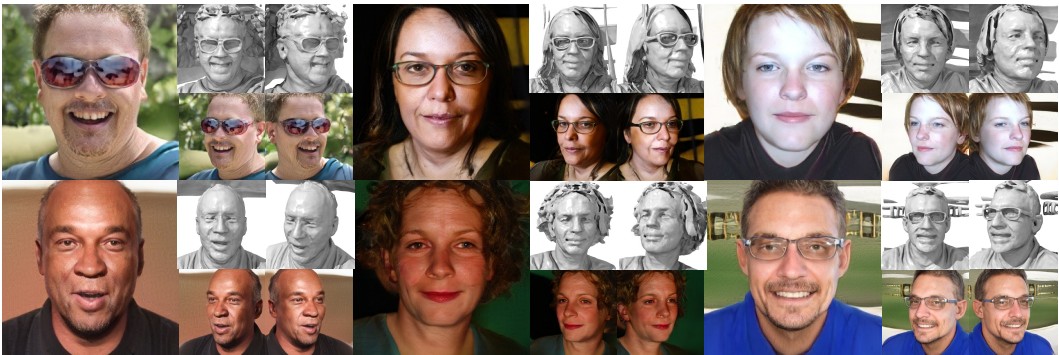

Figure A1: Samples synthesized on FFHQ [26] with truncation 0.7 using the model with an MLP-based point embedder. For each generated identity, we show the underlying geometry under two views and appearance under three views.

## C  More results

### C.1  Efficiency comparison

We report the training time and inference speed of models utilizing various point embedders in Tab. A1. The training time is computed by training our models on 8 NVIDIA A100 GPUs for 25 million images. Inference speed is measured on a single NVIDIA A100 GPU, where we processed 1K images and calculated the average FPS. As shown in Tab. A1, tri-plane-based point embedders exhibit superior computational efficiency compared to MLP-based and volume-based point embedders. MLP-based point embedders require a larger number of layers to extract point features, leading to longer processing times. Among these, volume-based point embedder is the least efficient, as it involves 3D convolutions.

### C.2  More qualitative results

We show more results of different point embedders on FFHQ [26] in Figs. A1 to A3. Interestingly, we observe that models equipped with tri-plane-based point embedders generate 3D shapes with sharper noses, while those equipped with MLP-based or volume-based point embedders do not exhibit this characteristic. This phenomenon can be observed more clearly in Fig. A4. However, the underlying reason for this remains unknown.

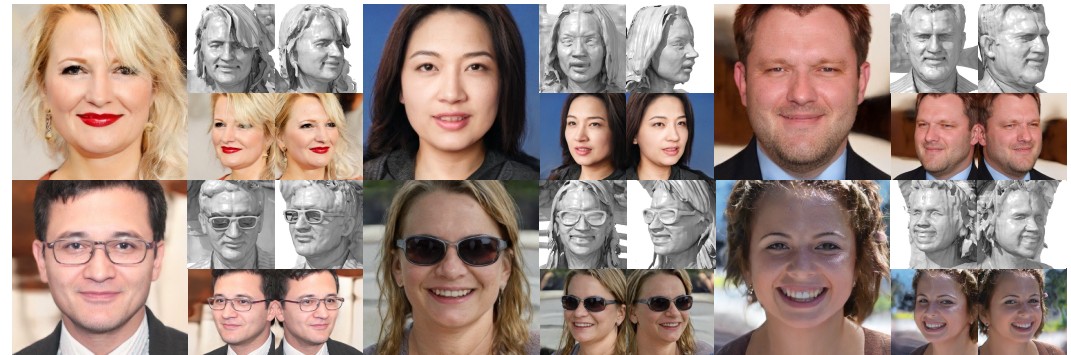

Figure A2: Samples synthesized on FFHQ [26] with truncation 0.7 using the model with a volume-based point embedder. For each generated identity, we show the underlying geometry under two views and appearance under three views.

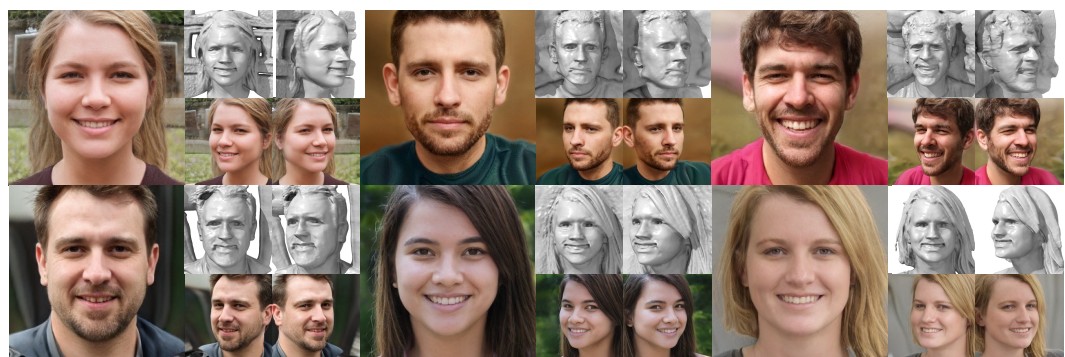

Figure A3: Samples synthesized on FFHQ [26] with truncation 0.7 using the model with a tri-plane-based point embedder. For each generated identity, we show the underlying geometry under two views and appearance under three views.

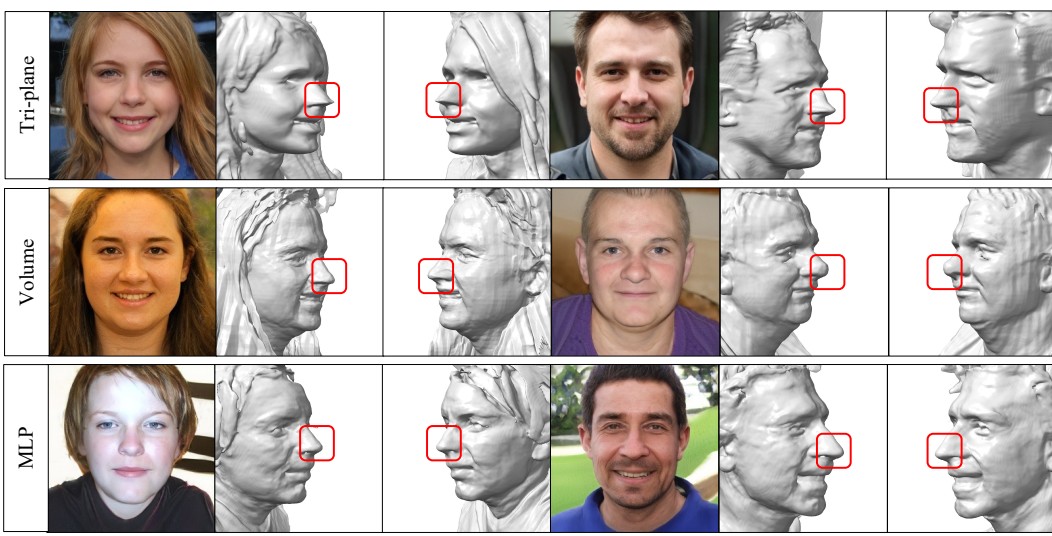

Figure A4: Qualitative comparison across different single point embedders on FFHQ [26], zoom in for better viewing. Models equipped with tri-plane-based point embedders generate 3D shapes with sharper noses, which can appear unnatural.

# D   Limitations and future work

**Efficiency.** Training our 3D GAN models is a time-consuming process, especially when utilizing MLP-based and volume-based point embedders. To draw meaningful conclusions, we must conduct a plethora of experiments. Currently we are uncertain about how to improve the efficiency of our models' training. Each experiment requires careful hyper-parameter tuning, which is a challenging task. However, due to computational limitations, multiple runs of our experiments are infeasible. Consequently, we identify the "optimal" settings using a limited number of attempts.

**Training stability.**   We have observed that training 3D GANs can be rather unstable. Some experiments are highly sensitive to hyper-parameters such as the $\gamma$ value of R1 regularization [34], learning rate, *etc.* However, our study does not investigate this aspect in depth. Future research addressing hyper-parameter sensitivity and training stability may lead to significant reductions in training costs and more compelling results.

**Universality.** We trained all our 3D-aware image synthesis models on simple, object-level datasets, such as FFHQ [26] for face generation, and our conclusions are based on these datasets. However, our paper does not explore the extension of 3D GANs to a higher degree of universality, which represents a promising research direction for the future. This universality pertains to generating diverse objects (*e.g.*, ImageNet [11] or Microsoft CoCo [31]), dynamic objects or scenes, and large-scale scenes.

