# OpenReview forum: "Benchmarking and Analyzing 3D-aware Image Synthesis with a Modularized Codebase"
_NeurIPS.cc/2023/Track/Datasets_and_Benchmarks — NeurIPS 2023 Datasets and Benchmarks Poster_

### Official Review · Reviewer_DaES · 2023-07-23

**Rating:** 7
**Confidence:** 3
**Correctness:** Yes
**Clarity:** Yes

**Strengths:**

> + Beneficial analyses: The analyses regarding different modules can help further work to build on top of a better baseline.
> + On-par or even better reproduction results: it is non-trivial to achieve on-par accuracy as compared to all baselines. Thus, this codebase can be one of the top options for beginners.
> + Modularized design space: the authors decoupled the complex design options in 3D-aware image synthesis into several modules, which enables the ability to allow users to replace any part with their proposed new modules.

**Additional Feedback:**

N/A

**Documentation:**

Yes

**Ethics:**

no ethical concern

**Limitations:**

The authors addressed the potential negative societal impact, but did not address the limitations.
See "Opportunities For Improvement" for the corresponding suggesttions.

**Opportunities For Improvement:**

> + Tutorial/User Guide to help beginners: Since the whole pipeline is designed to support different existing models, replacing existing modules with newly proposed ones is the most common use case. Thus, a well-written tutorial/user guide is important to enhance its visibility and better echo the claimed contribution of "our codebase allows users to replace a particular module".
> + Add more detailed discussions for future work and limitations: Although the authors discussed the future directions in Sec. 4.3, it is unclear whether they are critical bottlenecks for the current codebase. For instance, regarding hybrid representations, is it that the existing codebase supports this but lacks corresponding experimentation, or is it that the current code simply does not accommodate it?

**Relation To Prior Work:**

Yes

**Summary And Contributions:**

This work introduces a modularized codebase for 3D-aware image synthesis, which incorporates NeRF into the generator of GAN. The key contributions are: (1) a modularized easy-to-use codebase; (2) on-par or even better reproduction results, and (3) analyses regarding different modules.

---

> ### Author Response · Authors · 2023-08-21
> **Rebuttal by Authors**
>
> We sincerely thank the reviewer for the constructive reviews. Below are our responses addressing the concerns.
>
> ### Q1: Tutorial/User Guide.
>
> Thanks for the suggestion. We will prepare a README file with detailed instructions to help beginners familiar with our codebase. We will also prepare some example scripts to guide users in reproducing an existing method or combining techniques across different methods.
>
>
>
> ### Q2: More detailed discussions for future work and limitations.
>
> In the submission, our discussion primarily focuses on the limitations of existing 3D-aware image synthesis methods, which would be of interest to the entire community. It is noteworthy that we are able to identify these significant challenges within this domain mainly thanks to the analyses facilitated by our codebase. We present these analyses and discussions to inspire the research community, and hope that these challenges will be effectively addressed in future endeavors.
>
> As for the limitation of the proposed codebase, we need to point out its current form is built upon the pipeline of volumetric rendering. Even though this pipeline is popular and shared by all cutting-edge algorithms, it may still get replaced by some day, at which time our codebase would require a reimplementation. We will add the discussion.

---

### Official Review · Reviewer_M7ju · 2023-07-23
**This work is very helpful for the research community. Some improvement would make it brilliant.**

**Rating:** 7
**Confidence:** 5
**Correctness:** Mostly yes. Please check above sectio…
**Clarity:** Yes

**Strengths:**

1. This paper tries to make research clearer and easier by disentangling various factors in the implementation.
1. This paper provides useful observations for the research community. E.g., Table 1.
1. The background section thoroughly covers the literature.
1. Experiments cover diverse methods and ablations.

**Additional Feedback:**

.

**Documentation:**

n/a

**Ethics:**

No.

**Limitations:**

Yes L356

**Opportunities For Improvement:**

Opportunities for improvement
1. Identity consistency has a pitfall in that Arcface is not robust to pose changes. It should be noted.
1. CompCars is missing. It is a dataset containing real images of cars with complex background. It is more challenging than FFHQ, Cats, and ShapeNet Cars.
1. GIRAFFE and GIRAFFE-HD are missing.
1. Marching cube threshold is not covered.
1. Accurate pose distribution is not defined.
1. Pose conditioning is not covered.

Minor
1. uncertainty pose sampling -> stochastic pose sampling
1. FFHQ and Cats does not provide GT pose. They are estimated by off-the-shelf pose estimators. Please check the sentences rigorously.
1. Resolutions in Table 2 are different from L191.
1. Unnecessary abbreviations hurt readability. E.g., RDP, APD, GTP. random, accurate and GT would suffice.

**Relation To Prior Work:**

Modularization for analyses and reproduction.

**Summary And Contributions:**

This work provides a modularized codebase for 3D-aware GANs and performance analyses over different choices.

Performances are measured in FID, reprojection error, ID consistency, pose error, and depth error.

Core modules are pose sampler, point sampler, mapping network, point embedder, feature decoder, volume renderer and upsampler.

It also provides evaluator and visualizer.

---

> ### Author Response · Authors · 2023-08-21
> **Rebuttal by Authors**
>
> We sincerely thank the reviewer for the constructive reviews. Below are our responses addressing the concerns.
>
> ### Q1: Attention to ID metric.
>
> Thank you for your reminder. We will place greater emphasis on this limitation of ID metric in our revision.
>
> ### Q2: Experiments on CompCars.
>
> The CompCars dataset was not used by the majority of reference methods, including EG3D, EpiGRAF, StyleSDF, GRAM, and others. For those methods that have worked on this dataset (*e.g.*, StyleNeRF), the training is extremely unstable (see Issue 15 under the GitHub repository of StyleNeRF). Therefore, we do not consider it for benchmarking methods.
>
>
> ### Q3: GIRAFFE and GIRAFFE-HD.
>
> Following your suggestion, we support GIRAFFE in our codebase. The subsequent table showcases the comparative results between the official implementation and our reproduction.
>
> | GIRAFFE          |    Official     |   Reproduction    |
> |:----------------:|:---------------:|:-----------------:|
> | FFHQ 256x256     |     32.00       |      30.48        |
> | CompCars 256x256 |     26.00       |      26.67        |
>
> As for GIRAFFE-HD, we plan to incorporate it into our codebase in the future.
>
> ### Q4: Marching cube threshold.
>
> Thanks. The subsequent table reports the marching cube thresholds used for different datasets, which we will add in the revision.
>
> |Dataset     |Marching cube threshold|
> |:----------:|:---------------------:|
> |FFHQ        |           10          |
> |Cats        |           15          |
> |Cars        |           50          |
>
>
> ### Q5: Accurate pose distribution.
>
> We present the demonstration of accurate pose distribution in Section 1.6 of the supplementary material. Accurate pose distribution refers to the pose distribution of a dataset that has been pre-estimated using existing pose estimation methods. For example, for FFHQ dataset, we sample camera poses from a Gaussian distribution with the vertical standard deviation of 0.155 and the horizontal standard deviation of 0.3, which are estimated by the off-the-shelf pose estimators.
>
> ### Q6: Pose conditioning.
>
> This should be a misunderstanding. We have already incorporated pose conditioning related module into our framework, which is pose sampler. Our framework supports two types of pose conditioning: stochastic pose and deterministic pose. We transcript part of pose conditioning results (Table 9) of the submitted manuscript below.
>
> |Pose                        |      FID on FFHQ      |
> |:--------------------------:|:---------------------:|
> |MLP *w/* random pose        |           14.56       |
> |MLp *w/* accurate pose      |           9.96        |
> |MLP *w/* GT pose            |           5.15        |
> |Volume *w/* random pose     |           10.47       |
> |Volume *w/* accurate pose   |           7.34        |
> |Volume *w/* GT pose         |           4.65        |
> |Tri-plane *w/* random pose  |           15.18       |
> |Tri-plane *w/* accurate pose|           5.45        |
> |Tri-plane *w/* GT pose      |           4.72        |
>
> ### Q7: Some minor problems.
>
> 1. We will replace "uncertainty pose sampling" with "stochastic pose sampling" in our revision.
> 2. As the previous work considers poses from off-the-shelf estimators as ground-truth poses, we have decided to maintain the terminology used in our work. We will make it clear in our revision.
> 3. The resolution mentioned in L191 refers to the resolution used in our ablations, specifically in Section 4.2. Regarding Table 2, we primarily present the results based on the resolution consistent with the official implementation. We will make it clear in our revision.
> 4. Thank you for the valuable suggestion regarding the use of abbreviations. We will address this issue in our revision.

---

> > ### Comment · Reviewer_M7ju · 2023-08-28
> >
> > I have read the response. It resolves my concerns. I updated my rating from 6 to 7.
> >
> > FYI, GIRAFFE includes CompCars and it contradicts Q2. It is great to see CompCars being added even if it has only one method.

---

> > > ### Author Response · Authors · 2023-08-28
> > > **Response by Authors**
> > >
> > > Thanks for your constructive suggestions. We will carefully prepare the revision.

---

### Official Review · Reviewer_783m · 2023-07-27
**The paper is well-motivated and provide unified modularized codebase. Some weekness need to be addressed.**

**Rating:** 7
**Confidence:** 5
**Correctness:** Yes
**Clarity:** Yes

**Strengths:**

+ In general, it is a well-motivated paper and the writing is easy to follow. A modularized codebase is in need and would help boost the development of 3D-aware image synthesis.
+  The authors evaluate several representative methods and conduct detailed experiments on FFHQ.

**Additional Feedback:**

Please refer to the above.

**Documentation:**

Codebase is provided as a supplementary.  But, there was no specified maintenance plan regarding the project. it would be expected to know how the authors plan to support the codebase in the future, who will provide support, how to improve its effectiveness and flexibility, and any concrete play to include new algorithms. Moreover, as this codebase only support rigid object, is there any plan to extend to non-rigid (like human) ones?

**Ethics:**

No significant ethical concerns with current submission.

**Limitations:**

Please refer to *Opportunities For Improvement* and *Documentation*

**Opportunities For Improvement:**

+ （Could complement Tab2）How about the memory cost and training/test speed comparison of the database with each method's official implementation?
+  It seems like for point sampling, the authors only focus on pixel-space sampling (line124), without the flexibility of different sampling strategies along the ray. Please further discuss this point.
+  For Tab3, how do you combine the embedding from different Point Embedders? Would different strategies lead to different conclusions? In practice, how many strategies of feature fusion are supported in the codebase?
+ For the Geometric Representation (Tab7), it is expected to also evaluate on other datasets (cars, cat). For the cars, CD could be used as an additional metric.
+ The PE result in Tab7 is a bit counterintuitive, as with upsampler could achieve better pe. Please give further discussion to enhance the observations.

**Relation To Prior Work:**

N/A

**Summary And Contributions:**

This paper provides a codebase that modularizes the core components of current representative 3D-GAN methods (the current submission is focused on the rigid single object). The authors also conduct several ablations and conclude some suggestions on the usage of each individual component.

---

> ### Author Response · Authors · 2023-08-21
> **Rebuttal by Authors (Part 1)**
>
> We sincerely thank the reviewer for the constructive reviews. Please find below our responses addressing the raised concerns.
>
> ### Q1: The memory cost and test speed comparison.
>
> The table below presents a comparison between our codebase and the official implementation of each method, which is based on GPU memory cost and inference speed on the FFHQ dataset during inference:
> |Method   | Resolution| GPU memory (official) |  GPU memory (ours) | Inference speed (official) | Inference speed (ours) |
> |:-------:|:---------:|:---------:|:---------:|:---------:|:--------:|
> |GRAF     |  128x128  |  3.5 (GB) |  3.5 (GB) |   6 (fps) |  6 (fps) |
> |&pi;-GAN |  128x128  |  2.0 (GB) |  2.1 (GB) |   7 (fps) |  7 (fps) |
> |StyleSDF |  512x512  |  2.9 (GB) |  2.9 (GB) |  24 (fps) | 24 (fps) |
> |StyleNeRF|  512x512  | 2.0  (GB) | 2.1  (GB) |  25 (fps) | 25 (fps) |
> |VolumeGAN|  256x256  | 2.0  (GB) | 2.2  (GB) |  52 (fps) | 50 (fps) |
> |GRAM     |  256x256  | 13.5 (GB) | 14.2 (GB) |  3  (fps) |  3 (fps) |
> |EpiGRAF  |  256x256  | 12.2 (GB) | 12.4 (GB) |  14 (fps) | 13 (fps) |
> |EG3D     |  512x512  | 2.8  (GB) | 2.8  (GB) |  35 (fps) | 37 (fps) |
>
> Please note that all results were obtained by testing on a single NVIDIA A6000 GPU with a batch size of 1. For GRAF, &pi;-GAN, GRAM, and EpiGRAF, these models do not include an upsampler, resulting in the need to sample more points during inference. Consequently, NeRF rendering requires more computing resources, leading to higher GPU memory cost and reduced inference speed.
>
> ### Q2: Pixel-space point sampling and different sampling strategies along the ray.
>
> Pixel-space point sampling is based on the rendering resolution and generally adopts the uniform sampling strategy on the *image plane*. Rays are determined with the help of pixel-space sampled points and the camera intrinsics. We support multiple point sampling strategies *along the ray*, such as uniform sampling and hierachical sampling. Self-defined point sampling strategy can also be easily incorporated, benefiting from our highly-modularized framework.
>
>
> ### Q3: Combination of embeddings from different Point Embedders.
>
> For Table 3, we combine the embeddings from different Point Embedders by concatenating them along the feature channel dimension, which is illustrated in L50-51 of the spplementary material. We have also conducted experiments by adding the embeddings rather than concatenating. As shown in the table below, the impact of different strategies on feature fusion is not significant. We will add the discussions on feature fusion in the revised version.
>
> |Combination    |  FID  |   ID  |   DE   |     PE    |   RE    |
> |:-------------:|:-----:|:-----:|:------:|:---------:|:-------:|
> |Concatenation  | 4.62  | 0.769 |  0.467 |4.7$e^{-4}$|  0.091  |
> |Addition       | 4.69  | 0.758 |  0.441 |4.8$e^{-4}$|  0.098  |
>
>
> ### Q4: Geometric Representation experiments on Cars and Cats.
>
> We have conducted additional experiments of geometric representation on cars and cats. We report `FID` results in the following table.
>
> |Gemetric Representation |           Cats        |         Cars         |
> |:----------------------:|:---------------------:|:--------------------:|
> |SDF (MLP)               |          8.22         |        5.73          |
> |Density (MLP)           |          4.05         |        2.42          |
> |SDF (Volume)            |          5.10         |        5.52          |
> |Density (Volume)        |          3.59         |        2.25          |
> |SDF (Tri-plane)         |          10.71        |        5.88          |
> |Density (Tri-plane)     |          3.99         |        2.75          |
>
> We appreciate your suggestion regarding the CD metric for assessing the geometric quality of cars. Indeed, it is a valuable metric to consider. Given the constraints of the rebuttal period, we acknowledge the importance of incorporating this metric into our codebase and will add it for future evaluations.

---

> > ### Author Response · Authors · 2023-08-21
> > **Rebuttal by Authors (Part 2)**
> >
> > ### Q5: Counterintuitive PE result in Tab6.
> >
> > Please note that in Tab6, the PE results are presented in different resolutions (**256x256** for w/ upsampler and **64x64** for w/o usampler). To ensure a fair comparison, we conducted additional experiments on the 64x64 resolution of FFHQ dataset with an upsampler based on Tab6. The subsequent table presents the PE results for both with and without the upsampler at the same resolution. The results in the table support the conclusion that utilizing the upsampler leads to improved PE performance.
> >
> > |Activation Type         |             PE              |
> > |:----------------------:|:---------------------------:|
> > |SIREN(*w/* upsampler)     |         6.3$e^{-4}$     |
> > |SIREN(*w/o* upsampler)    |         6.6$e^{-4}$    |
> > |ReLU(*w/* upsampler)      |         4.5$e^{-4}$      |
> > |ReLU(*w/o* upsampler)     |         4.6$e^{-4}$         |
> >
> > ### Q6: Maintenance plan and extension to non-rigid objects.
> >
> > Thanks for the reminder. We will release the codebase to the entire community (maybe through GitHub), and we would like all researchers working in this field to help revise the implementation, support more algorithms, and develop new techniques together (maybe through the way of Pulling Request). Our team will serve as the primary members to help organize the development. We believe that the operation mode of `MMDetection` could play as a good reference.
> >
> > As for the extension to non-rigid objects, we would like to argue that this issue is more related to the methodology perspective (instead of the codebase perspective), as most existing algorithms (*e.g.*, EG3D) still perform poorly on non-rigid objects like human. One can freely use our codebase to learn a 3D-aware image synthesis model on human datasets, however, the peformance bottleneck is the algorithm itself but not our implementation.

---

> > > ### Comment · Reviewer_783m · 2023-08-28
> > >
> > > I changed my rating from 6 to 7, as the authors addressed most parts of my concerns. However, the feasibility of the maintenance plan is still a bit abstract. It is also unclear that whether the author team can in practice guarantee a stable, and long-term development of this codebase like OpenMMlab. After all, except for the community contribution, there is an entire department of the OpenMMLab team to support their toolbox maintenance. I would like to see the authors provide a more concrete discussion/plan in the revised version to ease the concern.
> > >
> > > BTW, I hoped to see a discussion on aspects like module flexibility on adding priors/deformation when talking about extending the codebase to non-rigid methods. I don't like this issue is more related to the pure methodology perspective. It will be great to see the authors add the discussion in the revised version.

---

> > > > ### Author Response · Authors · 2023-08-28
> > > > **Response by Authors**
> > > >
> > > > Thanks for your constructive suggestions. We will carefully prepare the revision.

---

### Official Review · Reviewer_ggV7 · 2023-07-27
**The motivation of the proposed modularized codebase is good, believe it can benefit image synthesis**

**Rating:** 7
**Confidence:** 3
**Clarity:** Writtting is good

**Strengths:**

The paper provides a highly modularized codebase for 3D-aware image synthesis, including a pose sampler, point sampler, stochasticity mapper, point embedder, feature decoder, volume renderer, upsampler, evaluator, and visualizer.
The proposed codebase contributes to re-implement a range of classic algorithms, and customize algorithms by replacing particular modules from the function perspective. This codebase allows users to change the module arbitrarily and independently for convenient algorithm development. The results for reproduced methods show that the codebase is able to generate equivalent performance. And the analysis of each module demonstrates that the code base is advanced in the capacity of the functionalities.

**Additional Feedback:**

no

**Correctness:**

This paper builds a "highly-modularized" "easy-to-use" codebase for 3D-aware image synthesis, but it seems to lack sufficient proofs to support this point. Comparisons between codebases and specialized metrics to compare different codebases are anticipated.

**Documentation:**

Documentation is sufficient

**Ethics:**

No ethical concerns

**Limitations:**

1.If the propsoed codebase is not the first in this field, the authors should introduce and compare related codebases, and give comparisons between them in the experiments. This is to show the advantages of this new work.
2. How to define "codebase" should be clearly given, is it a module-level library? but it also describes each single reference approach (e.g.,[7, 20, 53, 9]) as codebases, it may be confusing.

**Opportunities For Improvement:**

Since the codebase is developed and the modules are analyzed, a final optimal model can be integrated and demonstrated, and the performance should be compared with the existing methods. Meanwhile, the comparisons between the proposed codebase and existing codebases are not indicated sufficiently.

**Relation To Prior Work:**

it is clearly discussed

**Summary And Contributions:**

The paper builds a codebase for 3D-aware image synthesis by modularizing the generation process to incorporate a neural radiance field (NeRF) and a generative adversarial network (GAN). The main contribution of this paper is building a highly modularized codebase for 3D-aware image synthesis, which is beyond the capacity of the previous functionally entangled codebase.

---

> ### Author Response · Authors · 2023-08-21
> **Rebuttal by Authors**
>
> We sincerely thank the reviewer for the constructive reviews. Below are our responses addressing the concerns.
>
> ### Q1: Integrating and demonstrating a final optimal model.
>
> Recall that the primary objective of this work is to assist the community with an easy-to-use tool for 3D-aware image synthesis, such that users can develop their own algorithms with minor efforts. To demonstrate the practicality of our codebase, we show that
> - (correctness) One can achieve on-par or even better performance regarding the reproduction of existing method.
> - (flexibility) One can easily conbine existing methods (*e.g.*, use the combination of MLP, volume, and tri-plane to represent a spatial point) thanks to the modularized design of our codebase
> - (value to further research) One can analyze the effectiveness of each module in the popular pipeline of 3D-aware image synthesis, and identify some directions that worth further exploration.
>
> It is indeed possible to use our codebase to demonstrate a model with better performance than all existing approaches. Below we transcript some results from the submitted manuiscript to verify this point. However, we need to point out that these results are  achieved by simply combining existing techniques instead of proposing new techniques, and that these is *not* a general setting optimal for all datasets. We hope that our codebase could reduce the development difficulty of this field and inspire more studies for fundamental and general improvements. We will involve the above discussion to clarify the scope of this work.
>
> |              |   FFHQ 256x256  |   Cats 256x256   |   Cars 128x128  |
> |:------------:|:---------------:|:----------------:|:---------------:|
> |SOTA          |   4.80 (EG3D)   |    3.88 (EG3D)   |    2.75 (EG3D)  |
> |Best of ours  |     **4.62**    |     **3.59**     |    **2.25**     |
>
>
>
> ### Q2: About the definition of "codebase" and the comparison against existing codebases.
>
> Thanks. You are correct that our "codebase" refers to a module-level library, which can be directly applied to most classic algorithms for 3D-aware image synthesis. To the best of our knowledge, there currently lacks such a library in the community, which is the reason why we do not compare our codebase with existing codebases. We agree with you that calling existing single reference approach as "codebases" may indeed cause misunderstandings. They are more like a repository with the implementation of a particular method, leaving them difficult to borrow merits from other approaches. However, due to the lack of a well-designed codebase (or say, a library that can be shared across all methods), researchers currently choose to develop upon a single implementation. That is why we believe that our work is essential and would be of great interest to the communnity.
>
>
>
> ### Q3: Proofs to "highly-modularized" and "easy-to-use".
>
> We modularize the generation process into `pose sampler`, `stochasticity mapper`, `point sampler`, `point embedder`, `feature decoder`, `volume renderer` and `upsampler`, while in those individual implementations of prior works, some of the modules are entangled in different ways and thus are difficult to integrate into another work easily. With our highly-modularized codebase, arbitrary combinations of different techniques in different modules are allowed.

---

> > ### Comment · Reviewer_ggV7 · 2023-08-30
> >
> > A convincing response, rating upgraded.

---

### Author Response · Authors · 2023-08-21
**Author Response to ALL**

Dear Reviewers, ACs and PCs,

We sincerely thank the time and efforts invested by all the reviewers in reviewing our paper. Their valuable feedback has significantly contributed to enhancing the quality of our manuscript. We are encouraged to see that the reviewers appreciated the following aspects of our work:

* Building a highly modularized codebase for 3D-aware image synthesis [Reviewer `ggV7`, `783m`, `M7ju`, `DaES`]
* Observations and analyses are insightful and beneficial [Reviewer `ggV7`, `M7ju`, `DaES`]
* Reproduced results are on-par with, or even surpass, the original papers [Reviewer `ggV7`, `DaES`]
* Well-motivated paper [Reviewer `783m`]
* Writing is easy to follow [Reviewer `783m`]
* Codebase is easy to use [Reviewer `ggV7`, `DaES`]

We have addressed concerns raised by the reviewers point by point. Here, we would like to highlight some additional results suggested by the reviewers to confirm the **practicality** of our codebase.

- **Supporting GIRAFFE for 3D-aware image synthesis.**

| GIRAFFE          |    Official     |   Reproduction    |
|:----------------:|:---------------:|:-----------------:|
| FFHQ 256x256     |     32.00       |      30.48        |
| CompCars 256x256 |     26.00       |      26.67        |


- **More analyses of geometric representation on Cats and Cars datasets.**

|Geometric Representation |           Cats        |         Cars         |
|:----------------------:|:---------------------:|:--------------------:|
|SDF (MLP)               |          8.22         |        5.73          |
|Density (MLP)           |          4.05         |        2.42          |
|SDF (Volume)            |          5.10         |        5.52          |
|Density (Volume)        |          3.59         |        2.25          |
|SDF (Tri-plane)         |          10.71        |        5.88          |
|Density (Tri-plane)     |          3.99         |        2.75          |

We remain open to further discussions and welcome any additional comments from the reviewers.

Sincerely yours,

Authors

---

### Decision · Program_Chairs · 2023-09-22

**Decision:**

Accept (Poster)

**Comment:**

**Abstract:**
The paper addresses the advancement of 3D-aware image synthesis by creating a modularized codebase that integrates the neural radiance field (NeRF) into the generator of a generative adversarial network (GAN). The codebase aims to clarify the contribution of various techniques and offer a structured platform for researchers to fairly compare methods and develop modules independently. The codebase reproduces multiple advanced algorithms and provides in-depth analyses to deepen the understanding of existing methodologies.

**General Overview:**
Across all reviewers, there is a unanimous consensus that the paper is of value and should be accepted. The primary highlights recognized by reviewers include the establishment of a highly modularized codebase and its significant potential to accelerate the research and development in the 3D-aware image synthesis domain.

**Specific Highlights:**
  1. Modular Codebase Design: As pointed out by Reviewer ggV7, the paper's primary contribution is the development of a modularized codebase, encompassing several crucial components like pose sampler, point sampler, and volume renderer, to name a few. The design empowers users to modify or replace modules individually, making algorithm development more flexible and accessible.
  2. Reproduction and Performance: Reviewer ggV7 praises the fact that the introduced codebase is adept at re-implementing a diverse array of algorithms and maintains the performance of these reproduced methods. This showcases the codebase's capability to stand as a standard benchmark in the field.
  3. Contribution to the Research Community: Reviewer M7ju acknowledges the paper's attempt to simplify research by presenting a clearer picture of the multifaceted implementation aspects. The paper, through its observations (like those in Table 1), is regarded as a significant contribution to the wider research community.
  4. Background and Experiments: M7ju also commends the paper's comprehensive coverage of related literature, ensuring readers have a well-rounded understanding of the topic. The variety and depth of experiments, which include diverse methods and ablations, further strengthens the paper's findings.
  5. Clarity and Motivation: Reviewer 783m finds the paper's motivation clear and the writing coherent, which enhances its accessibility to a wider audience. There's also an acknowledgment of the demand for such a modularized codebase in the field, emphasizing its timeliness and relevance.

**Conclusion:**
The paper is well-received by the reviewers, with all of them recommending its acceptance. The modularized codebase's design and its potential to standardize research in 3D-aware image synthesis is particularly noteworthy. Furthermore, the paper's experiments, analysis, and literature coverage make it a valuable resource for the community. The consensus is that the paper fills a pertinent gap in the research arena, making it a commendable piece of work.